# GSLAMOT: A Tracklet and Query Graph-based Simultaneous Locating, Mapping, and Multiple Object Tracking System

## ABSTRACT

For interacting with mobile objects in unfamiliar environments, simultaneously locating, mapping, and tracking the 3D poses of multiple objects are crucially required. This paper proposes a Tracklet and Query Graph-based framework, i.e., GSLAMOT, to address this challenge. GSLAMOT utilizes camera and LiDAR multimodal information as inputs and represents the dynamic scene by a combination of *semantic map, agent trajectory, and an online maintained Tracklet Graph (TG)*. TG tracks and predicts the 3D poses of the detected active objects. A Query Graph (QG) is constructed in each frame by object detection to query and update TG, the semantic map, and the agent trajectory. For accurate object association, a Multi-criteria Star Graph Association (MSGA) method is proposed to find matched objects between the detections in QG and the predicted tracklets in TG. Then, an Object-centric Graph Optimization (OGO) method is proposed to simultaneously optimize the TG, semantic map, and the agent trajectory. It triangulates the detected objects into the map to enrich the map's semantic information. We address the efficiency issues to handle the three tightly coupled tasks in parallel. Experiments are conducted on KITTI, Waymo, and an emulated Traffic Congestion dataset that highlights challenging scenarios. Experiments show that GSLAMOT enables accurate crowded object tracking while conducting SLAM accurately in challenging scenarios, demonstrating more excellent performances than the state-of-the-art methods.

## CCS CONCEPTS

• **Computer systems organization** → **Robotic autonomy**; • **Computing methodologies** → *Simulation evaluation*.

## KEYWORDS

localization, 3D tracking, optimization, graph matching

## 1 INTRODUCTION

Conducting self-locating, mapping, i.e., SLAM, and multiple object 3D pose tracking (3D MOT) simultaneously using multimodel information is a fundamental requirement for dynamic scene perception, such as object tracking in autonomous driving, unmanned aerial vehicles (UAVs), and robotics human-machine collaboration [17, 22, 38].

**Unpublished working draft. Not for distribution.**

Multiple factors work collectively making this problem very challenging. (1) SLAM and 3D MOT need to be conducted concurrently and depend on each other. SLAM relies on object detection to eliminate the impacts of the dynamic objects for accurately tracking the agent poses and for static mapping, whereas 3D MOT relies on the accurate pose of the agent for triangulating the 3D poses of the mobile objects; (2) The concurrent movements of both the agent and the dynamic objects pose challenges for locating and object tracking; (3) Errors in possible object detection algorithms and sensor noise also contribute to inaccuracies in localizing surrounding objects; (4) Factors such as occlusions and high-speed movement also introduce difficulty in object matching, leading to locating and tracking errors.

Existing works in 3D multiple object tracking (3D MOT)[25, 34, 46] often assume that the ego-motion is known and free from noise or the sensor is fixed in the world frame. However, in real-world applications, it is general that both the agent and surrounding objects are in motion, requiring concurrent SLAM and 3D MOT. Furthermore, due to motion, the agent may produce inconsistent observations of the same object in different frames, leading to inaccuracies of the bounding boxes (Figure 3) and the difficulty in object matching across frames. Object-Oriented SLAM (OOSLAM) [3, 28, 33, 48] is another closely related area, but OOSLAM primarily focuses on estimating the ego-motion trajectory. It does not explicitly address the optimization of the detailed object 3D trajectories. But many applications do require real-time tracking of the 3D object poses[11, 18, 21].

To address the above challenges, this paper presents GSLAMOT, a graph matching and graph optimization based system for conducting SLAM and 3D MOT simultaneously, which takes stereo images and LiDAR point cloud sequences as input. To the best of our knowledge, it is the first work that outputs ego trajectory and object trajectories concurrently and accurately, even in challenging scenarios with crowded and highly dynamic objects. In particular, GSLAMOT represents the dynamic scene by a combination of *(1) a semantic map representing the environment, (2) the agent trajectory, and (3) an online maintained Tracklet Graph representing the 3D MOT trajectories*. TG tracks the 3D poses of multiple objects and can predict their poses at time $t$. A Query Graph (QG) is constructed in each frame based on object detection to query the predicted object poses. For accurate object association, a novel Multi-criteria Star Graph Association (MSGA) method is proposed for robust association to deal with the matching challenges in dynamic, congested, and noisy environments. MSGA evaluates neighborhood consistency, spatial consistency, and shape consistency between TG and QG, significantly improving multi-object tracking compared to using only spatial features.

After finding associations, an Object-centric Graph Optimization (OGO) method is proposed to simultaneously optimize the TG, the map, and the agent trajectory. We divide the optimization

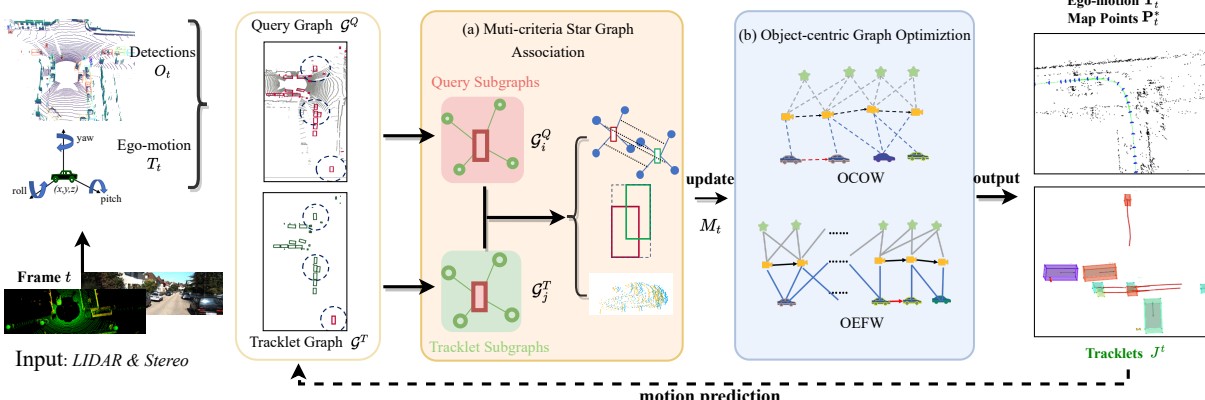

**Figure 1: Our system receives LiDAR point clouds and stereo images as inputs. The 3D detection algorithm extracts detection boxes from the point cloud, and the front-end of the visual odometry obtains the initial ego-motion pose. In the world frame, we construct query graphs and tracklet graphs for detections and tracklets, respectively. Then, we use the MSGA algorithm to perform association and tracking. Ego-motion, map points, and tracklets are optimized in OGO. The states of the tracklets in the next frame will be estimated through the motion model and then participate in the tracking of the next frame.**

into two parts: (1) a real-time Object-Centric Optimization Window (OCOW) and (2) a long-term Object-Ego Fusion Window (OEFW). In OCOW, a two-stage optimization strategy is adopted. Then, the well-optimized ego-motion, environmental points, and tracklets will be fused in the OEFW in a tightly coupled manner. Experiments show this object-centric graph optimization achieves better convergence speed (Figure 4) and accuracy than the ego-centric methods.

In system implementation, to enable the localization, 3D MOT, and semantic mapping to run concurrently and efficiently, We employ multi-thread parallel, executing modules such as visual front-end, mapping, detection, feature extraction, and tracking concurrently across threads. The main contributions of this work are as follows:

- We present GSLAMOT, which utilizes multimodel information, including stereo images and LiDAR point clouds, as inputs, and uses graph matching and graph optimization to conduct SLAM and 3D MOT simultaneously even in challenging scenarios.
- Multi-criteria Star Graph Association (MSGA) is proposed to match QG and the predicted TG.
- Object-Centric Graph Optimization (OGO) is proposed to estimate tracklets' poses. Object-Ego Fusion Window (OEFW) is proposed to jointly optimize the object poses and the ego-motion poses in a long-term sliding window.
- We exploit parallel threads to address the concurrency requirements in implementing the GSLAMOT system. Experiments demonstrate that GSLAMOT exhibits better real-time performance than the OOSLAM systems without object tracking.
- We have also created a Traffic Congestion dataset for object-level scene understanding using the Carla simulator[9]. The dataset encompasses a wide range of maps and has a varying

number of dynamic and static objects, providing a valuable resource for research and development in this field.

## 2 RELATED WORK

The most closely related works are reviewed, which are categorized into two areas: Object-Oriented SLAM (OOSLAM) and 3D Multiple Object Tracking (3D MOT).

### 2.1 Object-oriented SLAM

Early semantic SLAM systems typically focus on removing dynamic objects to reduce the influence of dynamic features on pose estimation, including DynaSLAM[3], BlitzSLAM[10], and DynaVINS[30]. However, these systems do not provide poses of dynamic objects.

Object-oriented SLAM (OOSLAM), as an important branch in the research of object-level scene understanding, aims to extract, model, and track the static and dynamic objects in the environment[1]. Early OOSLAM systems[28, 32] typically maintain a database of objects. These systems utilize RGB-D cameras as inputs and employ ICP losses along with pose graph optimization to solve for the camera and object poses. However, this model-based approach requires prepared object models in advance and cannot exhibit robust scalability to unknown scenes. Recent advancements in OOSLAM have increasingly focused on improving object feature extraction and modeling. In QuadricSLAM[24], objects are modeled as ellipsoids or quadric geometry, enabling more effective optimization of shape and scale. CubeSLAM[36] aligns the detection bounding boxes with the image edges to enhance object proposals. DSP-SLAM[33] leverages the signed distance function (SDF) to reconstruct the objects after detection. The learned shape embedding serve as priors during the reconstruction, and the shapes of the objects are jointly optimized in the bundle adjustment process. In MOTSLAM[41], Zhang et al. apply three neural networks to extract 2D bounding boxes, 3D bounding boxes, and semantic segmentation, respectively.

While the multi-network approach can extract accurate information about an object from different scales, it does demand large computational resources. DynaSLAM II[2] employs instance semantic segmentation and tracks ORB features on dynamic objects. Similarly, VDO-SLAM[42] leverages instance semantic segmentation, utilizing dense optical flow to maximize the number of tracked points on moving objects. Furthermore, dense optical flow is applied to ensure consistent tracking across multiple objects.

However, most of OOSLAM systems only utilize objects as landmarks for observation, lacking a complete multi-object tracking process and failing to output trajectories for surrounding objects. For the noises in object detection and camera pose tracking, object tracking remains a challenging problem in SLAM settings, especially in object-crowded environments.

## 2.2 3D Multiple Object Tracking

2D multi-object tracking, which involves tracking object bounding boxes in the image plane, has been extensively studied [40, 43–45]. 3D multi-object tracking remains a challenging problem due to the involvement of spatial motion and the 3D appearance of the objects [26, 29].

The framework of 3D MOT is mainly divided into tracking by detection and learning-based algorithms. The former mainly uses Kalman filtering to estimate objects and then relies on specific indicators such as Intersection of Union (IoU)[34], Mahalabnobis distance[7], or Generalized IoU[25] for association.

For learning-based algorithms, many works model 3D MOT as GNN, representing the association information of objects by predicting edges[4, 35, 39]. More relevant to this paper is the approach based on geometric information. PolarMOT[16] only uses 3D boxes as inputs to GNN to learn the geometric features of objects. BOTT[46] relies on self-attention to represent global context information and associate boxes.

Existing 3D MOT approaches generally work either in the ego-motion frame or under the assumption of known ego-motion. However, SLAM and 3D MOT are specially required in unknown environments or in the presence of ego-motion noise. Therefore, our GSLAMOT presents a practical approach.

## 3 APPROACH OVERVIEW

The proposed GSLAMOT framework is shown in Figure 1. We consider the agent is conducting self-locating, environment mapping, and multiple object 3D pose tracking in an unfamiliar environment. The agent is equipped with a stereo camera that captures RGB image pairs, and a 3D LiDAR. The stereo images are processed by the visual odometry (VO) system and the LiDAR point clouds are used for 3D object detection[19, 20].

For the $t$-th frame, the sensor input is denoted as $\Omega_t$. The 3D detector provides the object detection results $O_t$. The VO front-end outputs the current ego-motion pose $T_t$. The detections $O_t$ are transformed into the world coordinate frame by $Q_t = T_t O_t$, and the Query Graph $\mathcal{G}^Q$ is constructed.

$$O_t = f_{detector}(\Omega_t) \tag{1}$$

$$T_t = f_{odometry}(\Omega_t, T_{t-1,...}) \tag{2}$$

$$\mathcal{G}^Q = f_{qconstruct}(Q_t) \tag{3}$$

In processing this new frame, the active tracklets $\mathcal{J}_{t-1}$ are predicted to time $t$ based on motion model, denoted as $\mathcal{J}_t^{pred}$. Then a tracklet graph $\mathcal{G}^T$ is constructed based on $\mathcal{J}_t^{pred}$. Subsequently, we employ Multi-criteria Star Graph Association (MSGA) to match between $\mathcal{G}^Q$ and $\mathcal{G}^T$ to obtain the matching relationship $\mathcal{M}_t$.

$$\mathcal{J}_t^{pred} = f_{motion}(\mathcal{J}_{t-1}) \tag{4}$$

$$\mathcal{G}^T = f_{tconstruct}(\mathcal{J}_t^{pred}) \tag{5}$$

$$\mathcal{M}_t = f_{MSGA}(\mathcal{G}^Q, \mathcal{G}^T) \tag{6}$$

Then, We employ Object-centric Graph Optimization (OGO) to optimize the ego-motion $T_t$, tracklets $\mathcal{J}_t$, and the map points $P_t$. OGO operates within a window size $w$, incorporating Object-Centric Optimization Window (OCOW) and Object-Ego Fusion Window (OEFW). The optimized results are used to update the Tracklet Graphs.

$$T_t, \mathcal{J}_t, P_t = f_{OCOW}(T_t, \mathcal{M}_t, \mathcal{J}_t^{pred}, Q_t) \tag{7}$$

$$T_{t-w:t}^*, \mathcal{J}_{t-w:t}^*, P_{t-w:t}^* = f_{OEFW}(T_{t-w:t}, \mathcal{M}_{t-w:t}, \mathcal{J}_{t-w:t}, Q_{t-w:t}) \tag{8}$$

We use Pointpillars[19] for LiDAR-based 3D object detection, and use ORB-SLAM3[5] for visual odometry front-end. Their results, i.e., $O_t$ and $T_t$ are inputs of following steps.

## 4 MULTI-CRITERIA STAR GRAPH ASSOCIATION

We firstly introduction the construction of TG and QG, and the Multi-criteria Star Graph Association (MSGA).

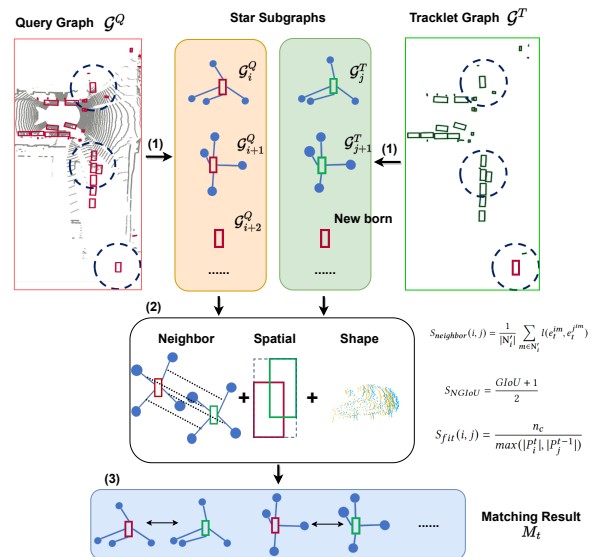

Figure 2: Multi-criteria Star Graph Association.

## 4.1 Query Graph and Tracklet Graph

At frame $t$, we select the active tracklets from frame $t - w$ to frame $t - 1$ as $\mathcal{J}_{t-1}$, where $w$ is the OGO window size (detailed in Section 5). The critical task at this stage is to correctly associate the objects

in $\mathbf{O}_t$ with those in $\mathcal{J}_{t-1}$. The association is challenging for the following reasons: (1) the agent and the objects are moving; (2) the objects may be crowded; and (3) the detections may be noisy.

Thus, we firstly predict the poses of the tracklets, i.e., $\mathcal{J}_t^{pred} = f_{motion}(\mathcal{J}_{t-1})$, where $f_{motion}$ is based on Kalman Filter[25] using the objects' estimated motion velocities and historical trajectories. Then we construct a graph for both the detections $\mathbf{O}_t$ and the predicted poses of the tracklets, named Query Graph (QG) and Tracklet Graph (TG) respectively.

In particular, in constructing QG, we firstly transform the poses of the detected 3D boxes to the world frame, obtaining $\mathbf{Q}_t$. Then for each detection $i$ in $\mathbf{Q}_t$, the nearest $K$ detections within distance $L$ are chosen as $i$'s neighbor set $\mathbf{N}_i$. We build edges between each detection and its neighbors. This converts the detection set $\mathbf{Q}_t$ to a graph $\mathcal{G}^Q$. Each detection $i$ also forms a local star-graph $\mathcal{G}_i^Q = (\mathcal{V}_i, \mathcal{E}_i)$, where $\mathcal{V}_i = \{i\} \cup \mathbf{N}_i$, and $\mathcal{E}_i$ includes the edges from $i$ to each element in $\mathbf{N}_i$.

In constructing TG, for each $j$ in $\mathcal{J}_t^{pred}$, the nearest $K$ predicted object poses within distance $L$ are chosen as $j$'s neighbor set $\mathbf{N}_j$. We also build edges between each tracklet and its neighbors, and converts $\mathcal{J}_t^{pred}$ to a graph $\mathcal{G}^T$. Each predicted object $j$ also forms a local star-graph $\mathcal{G}_j^T = (\mathcal{V}_j, \mathcal{E}_j)$, where $\mathcal{V}_j = \{j\} \cup \mathbf{N}_j$, and $\mathcal{E}_j$ includes the edges from $j$ to its neighbors.

## 4.2 Multi-criteria Consistency

The purpose of association is to determine whether a detection $i \in \mathbf{Q}^t$ and a tracklet $j \in \mathcal{J}_t^{pred}$ are corresponding to the same object in the environment. We access this by multi-criteria of $\mathcal{G}_i^Q$ and $\mathcal{G}_j^T$. Two subgraphs with similar structures and features should exhibit a high graph feature consistency[23]. We particularly consider consistency evalution from three aspects: (1) *neighborhood consistency*, which can be considered as edge similarity; (2) *spatial consistency*, evaluated by Normalized Generalized Intersection over Union (NGIoU)[27] and (3) *shape consistency*, evaluated by Iterative Closest Point (ICP)[15] between the point clouds in the object bounding boxes.

*4.2.1 Neighborhood Consistency.* In detection $i$'s local star graph, the edge $e^{im}$ between vertex $i$ and $m \in \mathbf{N}_i$ can be represented by a relative pose transformation from $i$ to the neighbor $m$, denoted by $\mathbf{T}_t^{im}$. If $i \in \mathcal{G}_i^Q$ and $j \in \mathcal{G}_j^T$ are consistent, the local neighborhood edges of $i$ and $j$ should be highly consistent. For example, if vertex $m$ in $\mathcal{G}_i^Q$ is corresponding to vertex $n$ in $\mathcal{G}_j^T$, $\mathbf{T}_t^{jn}$ should be consistent with $\mathbf{T}_t^{im}$. So we evaluate the neighborhood consistency between $e_t^{im}$ and $e_t^{jn}$ by:

$$l(e_t^{im}, e_t^{jn}) = exp(-\|\mathbf{T}_t^{im}(\mathbf{T}_t^{jn})^{-1} - \mathbf{I}\|_F) \qquad (9)$$

where $\mathbf{T}_t^{im}$ is the transformation of $e_t^{im}$ and $(\mathbf{T}_t^{jn})^{-1}$ is the inverse transformation of $e_t^{jn}$. $\|\cdot\|_F$ represents the Frobenius norm.

However, at this stage, we have not yet completed the object association, meaning that for the query and tracklet graphs, the correspondence of leaf vertices ($m$ and $n$ in Equation (9)) is unknown. This presents a classic chicken-and-egg problem. We propose a **Greedy Neighbor Strategy** to address this issue. For query local

star graph $\mathcal{G}_i^Q$ and tracklet local star graph $\mathcal{G}_j^T$. We firstly assume their central vertices represent the same object, then if the edge $e^{im}$ in $\mathcal{G}_i^Q$ exhibits the highest consistency (Equation 9) with the edge $e^{jn}$ in $\mathcal{G}_j^T$, $e^{im}$ and $e^{jn}$ are considered the corresponding edges.

If the highest consistency edge pairs $\mathcal{G}_i^Q$ and $\mathcal{G}_j^T$ do not actually correspond to each other, we still select the edge in $\mathcal{G}_j^T$ with the highest consistency for each edge in $\mathcal{G}_i^Q$, even if their consistency is low. This low consistency can contribute to discrimination in multi-criteria star graph matching (Section 4.3). The overall neighborhood consistency score between $\mathcal{G}_i^Q$ and $\mathcal{G}_j^T$ is therefore:

$$S_{neighbor}(i, j) = \frac{1}{|\mathbf{N}_i'|} \sum_{m \in \mathbf{N}_i'} l(e_t^{im}, e_t^{j^{im}}) \qquad (10)$$

where $\mathbf{N}_i'$ contains the leaf vertices in $\mathcal{G}_i^Q$ that have corresponding vertices in $\mathcal{G}_j^T$. $e_t^{j^{im}}$ represents the corresponding edge of $e_t^{im}$ in $\mathcal{G}_j^T$, which is obtained from the Greedy Neighbor Strategy.

*4.2.2 Spatial Consistency.* Traditional 3D object tracking typically relies on spatial metrics based on Intersection over Union (IoU)[34] or inter-distance[7]. However, traditional methods based on inter-distance or IoU between the current detection and tracklets may fail when the distance or IoU values exceed thresholds. This is generally when the agent or objects move at high speed or when there are detection errors. Furthermore, IoU cannot handle cases where two bounding boxes do not overlap, leading to redundant objects in the tracking results.

To address the issues above, we utilize Normalized Generalized Intersection over Union (NGIoU) as our spatial consistency metric. For two centering vertices $i$ and $j$ corresponding to two star graphs (for instance, detection $i$ and tracklet $j$), we suppose their 3D bounding boxes are $B_i$ and $B_j$ respectively. Then the value of GIoU is calculated as follows:

$$GIoU(i, j) = \frac{|B_i \cap B_j|}{|B_i \cup B_j|} - \frac{|B_i \cup B_j| - |B_i \cap B_j|}{|B_i \cup B_j|} \qquad (11)$$

We further normalize GIoU to ensure its range lies within $[0, 1]$:

$$S_{NGIoU} = \frac{GIoU + 1}{2} \qquad (12)$$

Experimental results in Section 7 demonstrate that this metric enhances object tracking performances in challenging scenarios.

*4.2.3 Shape Consistency.* Traditional 3D object tracking methods [7, 34] rarely leverage semantic information within the bounding boxes, while deep learning-based approaches rely on implicit features. We find that the point cloud within the bounding box inherently represents the shape of the object, which can be considered as semantic information and is an important clue for evaluating object consistency. Although the shapes of the same object in consecutive frames are similar, the view angles maybe different. So we propose a point cloud registration based methods to evaluate the point cloud shape similarity.

For detection $i$ in $\mathcal{G}^Q$ and tracklet $j$ in $\mathcal{G}^T$, the point clouds within their respective bounding boxes are denoted as $P_i^Q$ and $P_j^T$.

The ICP algorithm with RANSA[47] is used to compute the point-to-point correspondence between $P_i^Q$ and $P_j^T$. The fitness score calculated for the two point clouds is given by:

$$S_{fit}(i,j) = \frac{n_c}{max(|P_i^Q|, |P_j^T|)} \quad (13)$$

where $n_c$ represents the number of successfully registered points.

## 4.3 Subgraph Matching

For each $\mathcal{G}_i^Q$ and its central vertex $i$, we first collect the candidata set $\mathbf{C}_i$. For each tracklet star-graph in $\mathbf{C}_i$, the distance between its central vertex and $i$ is less than $L$. For a query star-graph $\mathcal{G}_i^Q$ and a tracklet star-graph $\mathcal{G}_j^T$ in $\mathbf{C}_i$, the multi-criteria feature consistency is given by:

$$S(i,j) = \lambda_1 S_{neighbor}(i,j) + \lambda_2 S_{NGIoU}(i,j) + \lambda_3 S_{fit}(i,j) \quad (14)$$

If $S(i,j)$ falls below a threshold $\tau$, then we consider $i$ and $j$ should not be associated, leading to the removal of $j$ from the candidate set $\mathbf{C}_i$. If the candidate set $\mathbf{C}_i$ is empty, this indicates that $i$ is a newly detected object does not originally exist in the map. We create a new tracklet, which is added as the only element in $\mathbf{C}_i$. Finally, we utilize the Kuhn-Munkres algorithm to compute the final one-to-one detection-to-tracklet matching by seeking maximum consistency:

$$\mathcal{M}_t^* = \arg\max_{\mathcal{M}_t \in \Psi_t} \sum_{i \in Q_t, j = \mathcal{M}_t^i} S(i,j) \quad (15)$$

where $\Psi_t$ represents the set of all possible matches and $j = \mathcal{M}_t^i$ represents the association of detection $i$ with tracklet $j$ under the matching relationship $\mathcal{M}_t$.

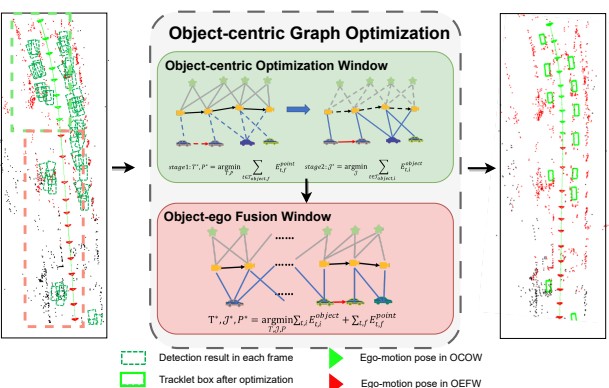

Figure 3: The Object-centric Graph Optimiaztion (OGO). Before optimization, the detection boxes for each object at different frames are inconsistent. After applying OGO, we obtain accurate object poses. The residual edges in solid lines participate in graph optimization operations, while the residual edges in dashed lines do not participate.

## 5 OBJECT-CENTRIC GRAPH OPTIMIZATION

Graph optimization is the widely employed back-end in SLAM (Simultaneous Localization and Mapping)[8, 13]. It represents ego-motion states and sensor measurements as nodes and edges in a graph, utilizing optimization algorithms to estimate the agent's trajectory and the environment map. However, we observed that this approach performs well primarily in static scenes.

For tracking dynamic objects, the ego-motion errors and the object pose errors coexist, affecting the convergence speed and accuracy of the graph optimization. To enable optimization tailored for 3D tracklets, we propose a graph optimization framework named Object-centric Graph Optimization (OGO). We divide the sliding window into two parts: Object-centric Optimization Window (OCOW) and Object-Ego Fusion Window (OEFW), and two windows adopt different optimization strategies.

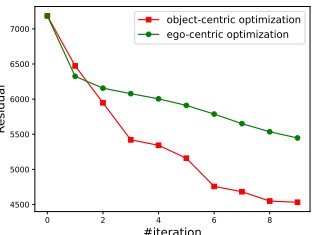

Figure 4: The residual curves of our proposed object-centric optimization and classic ego-centric optimization. The residual refers to the summary of all mapping-based errors and object detection errors in Equation (19). Our proposed OGO strategy can better optimize the system.

## 5.1 Object-centric Optimization Window

Before graph optimization, we obtain initial ego-motion poses and map points for each frame from the visual odometry. Additionally, initial tracklets and the matching result $\mathcal{M}_t$ for each frame are also available after tracking and association.

In the object-centric optimization window, we adopt a two-stage optimization strategy (Figure 3). In the first stage, we solely utilize residuals from static environment landmarks in SLAM and ego-motion poses to estimate ego-motion. At this point, the ego motion serves as a relatively reliable initial value. The objective function of the first stage is:

$$stage\ 1 : \mathbf{T}^*, \mathbf{P}^* = \arg\min_{\mathbf{T},\mathbf{P}} \sum_{t \in \mathcal{T}_{OCOW}} E_t^{map} \quad (16)$$

where $\mathcal{T}_{OCOW}$ represents the set of frames in OCOW and $E_t^{map}$ represents the mapping residuals in ORBSLAM3[5]. $\mathcal{T}_{OCOW}$ represents the set of ego-motion poses, and $\mathbf{P}$ represents the set of map point positions.

After the first stage, the estimated ego-motion is reliable due to the elimination of dynamic objects in estimation. Then, we fix the ego-motion and solely optimize the object poses using residuals from object detection. The object detection residual at frame $t$ is:

$$E_t^{object} = \sum_{i \in O_t} ||\mathbf{T}_t^{cw} \mathbf{T}_t^{wj} (\mathbf{T}_t^{ci})^{-1}||_F \quad (17)$$

where $j = \mathcal{M}_t^i$ and $\mathbf{T}_t^{ab}$ represents the transformation from $a$ to $b$ at frame $t$. The objective function for the sliding window is:

$$stage\ 2 : \mathcal{J}^* = \arg\min_{\mathcal{J}} \sum_{t \in \mathcal{T}_{OCOW}} E_t^{object} \quad (18)$$

## 5.2 Object-ego Fusion Window

We combine reliable tracklets and ego-motion poses for joint optimization in OEFW, thereby enhancing the accuracy of both.

When the number of frames where a static object is observed exceeds a threshold $w$, we move it from the object-centric optimization window (OCOW) to the object-ego fusion window (OEFW). Once all detections in a frame have been moved to the OEFW, we transfer that frame's ego-motion pose and map points to the OEFW. In OEFW, objects and tracklets have undergone sufficient multi-frame observations, possessing good initial values and low system error. Reliable observation and joint optimization can help correct cumulative errors and improve the accuracy of locating and tracking. We employ a tightly coupled optimization strategy within the window, jointly optimizing ego-motion poses, map points, and tracklet poses. The objective function of OEFW is:

$$\mathbf{T}^*, \mathcal{J}^*, \mathbf{P}^* = \underset{\mathbf{T}, \mathcal{J}, \mathbf{P}}{\arg\min} \sum_{t \in \mathcal{T}_{OEFW}} E_t^{object} + E_t^{map} \quad (19)$$

where $\mathbf{T}, \mathcal{J}, \mathbf{P}$ represent the set of ego-motion poses, the set of object poses, and the set of landmark positions, respectively. $\mathcal{T}_{OEFW}$ is the set of frames in OEFW.

## 6 SYSTEM IMPLEMENTATION DETAILS

### 6.1 System Acceleration

GSLAMOT demands high real-time performance for localization and tracking. To enhance system efficiency, we leverage parallel operations, effectively reducing the processing time for each frame of data. Specifically, tasks including visual front-end localization, environment map construction, object detection, and tracking are processed concurrently by different threads. The VO front-end determines whether a frame is a keyframe[5]. Keyframes are processed by MSGA and OGO. Localization and mapping for non-keyframes proceed in other parallel threads, thus eliminating the need to wait for time-consuming keyframe operations. Specifically, the odometry, mapping, detection, and submodules of OGO and MSGA all run on separate threads (Figure 5). In addition, we utilize parallel acceleration for point cloud operations and registration[47] during the computation of shape features. Experimental results demonstrate that our system achieves real-time performance and incurs less time overhead compared to other OOSLAM systems (Section 7.6).

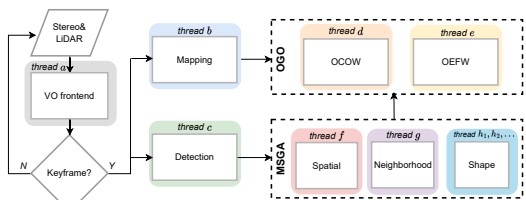

**Figure 5: The system architecture and multiple threads for parallel.**

### 6.2 Traffic Congestion Dataset

Highly dynamic and congested traffic environments are ubiquitous in the real world, but few data sets emphasize these scenarios. To address this challenge, we use the Carla simulator[9] to simulate dynamic and crowded scenes, and generate the **Traffic Congestion Dataset (TCD)**. We collected several sequences from four maps and the sensors on the vehicle include stereo cameras and 64-channel LiDAR data.

In addition to the ego vehicle, hundreds of other dynamic or static vehicles are on the map to simulate dynamic and crowded environments in reality. In each segment sequence of the Waymo Open Dataset[31], the average number of vehicles is less than one hundred, while in our Traffic Congestion dataset, the number of vehicles per segment sequence ranges from 200 to 300. The dense and dynamic flow of vehicles pose challenges to accurate self-localization and MOT. We will open-source this dataset to support further research in localization and 3D object tracking.

## 7 EXPERIMENTS

### 7.1 Experimental Settings

We perform evaluations of the proposed algorithm and compare it with the state-of-the-art algorithms, including the SLAM-based systems: ORBSLAM3[5], DynaSLAM[3], VDO-SLAM[42], DSP-SLAM[33] and MOT algorithms: AB3DMOT[34], ProbTrack[7], CenterPoint[37], BOTT[46], SimpleTrack[25], TrajectoryFormer[6]. All experiments are conducted on a computer with a CPU of i7-12700, GPU of RTX 3060Ti, and 16G RAM. We adopt ORB-SLAM3 as our visual odometry front-end. We use Pointpillars[19] for 3D object detection. We evaluated our system on the KITTI benchmark[12], Waymo Open Dataset[31] and our emulated Traffic Congestion dataset.

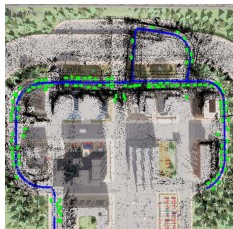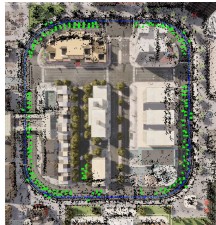

**Figure 6: The examples of the maps in Traffic Congestion dataset and the outputs of GSLAMOT. Blue points represent ego-motion trajectories, green boxes represent tracklets, and black points represent map points.**

### 7.2 3D Object Tracking Evaluation

We evaluate the results of 3D MOT on both the Traffic Congestion Dataset (TCD) (Table 2) and the widely used Waymo Open Dataset (WOD) (Table 1). For the WOD, we report the mismatch metrics and the MOTA metrics for each of the three categories (vehicle, pedestrian, and cyclist).

Researches on 3D MOT typically assume known ego-motion poses. However, our system computes the ego-motion in real-time, making the comparisons unfair. Therefore, we provide two system implementations: one calculates ego-motion poses simultaneously (GSLAMOT) and another utilizes ground truth ego-motion poses as inputs (GSLAMOT*). Results show that our methods outperform other methods in both known ego-motion and unknown ego-motion scenarios.

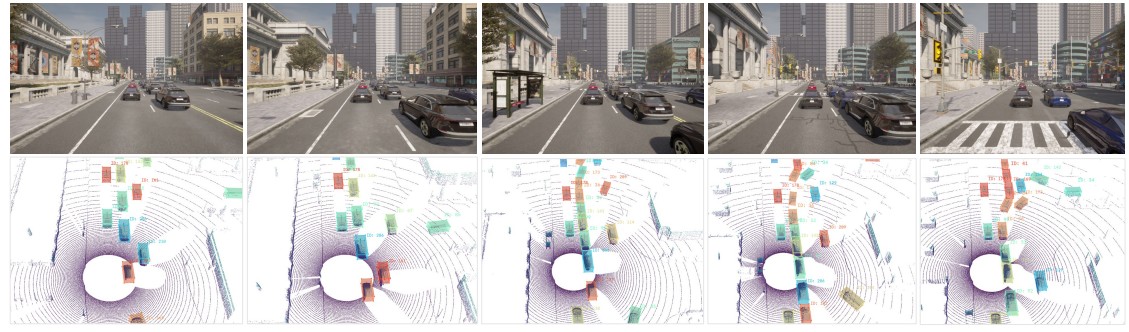

**Figure 7: Examples of 3D MOT on Traffic Congestion dataset.**

**Table 1: 3D MOT Evaluation on Waymo Open Dataset. The top one is in bold and the second is underlined.**

| Method | MOTA(L1)↑ | MOTA(L2)↑ | Mismatch↓ | MOTA(L2)↑ | | |
|---|---|---|---|---|---|---|
| | | | | vehicle | pedestrian | cyclist |
| AB3DMOT[34] | - | - | - | 40.1 | 33.7 | 50.39 |
| ProbTrack[7] | 48.26 | 45.25 | 1.05 | 54.06 | 48.10 | 22.98 |
| CenterPoint[37] | 58.35 | 55.81 | 0.74 | 59.38 | 56.64 | 60.0 |
| SimpleTrack[25] | 59.44 | 56.92 | 0.36 | 56.12 | 57.76 | 56.88 |
| BOTT[46] | 59.67 | 57.14 | 0.35 | 59.49 | 58.82 | 60.41 |
| TrajectoryF[6] | - | - | - | 59.7 | **61.0** | **60.6** |
| GSLAMOT | 59.69 | 57.10 | **0.33** | 60.45 | 60.02 | 60.33 |
| GSLAMOT* | **59.75** | **57.20** | **0.33** | **60.47** | 60.23 | 60.45 |

GSLAMOT: The ego-motion poses are estimated by odometry front-end.
GSLAMOT*: The groundtruth ego-motion poses are given as other MOT algorithms.

**Table 2: 3D MOT Evaluation on Traffic Congestion Dataset. The top one is in bold and the second is underlined.**

| | MOTA↑ | MOTP↑ | IDS↓ | Recall↑ | Precision↑ |
|---|---|---|---|---|---|
| AB3DMOT[34] | 36.95 | 39.42 | 99 | 36.49 | 37.05 |
| ProbTrack[7] | 39.07 | 45.56 | 77 | 47.23 | 43.87 |
| SimpleTrack[25] | 43.12 | 59.91 | 60 | 56.35 | 55.27 |
| GSLAMOT* | **54.36** | **70.92** | **15** | **68.05** | **66.89** |
| DSP-SLAM[33] | 20.95 | 51.58 | 51 | 30.01 | 57.21 |
| VDO-SLAM[42] | 33.24 | 53.1 | 29 | 40.68 | 55.12 |
| GSLAMOT | **49.10** | **69.71** | **22** | **67.55** | 65.52 |

GSLAMOT: The ego-motion poses are estimated by odometry front-end.
GSLAMOT*: The groundtruth ego-motion poses are given as other MOT algorithms.

## 7.3 Robustness Evaluation

The environment conditions, like rainy, snowy, and foggy days, may interfere with hardware devices and algorithms and then may produce noises. However, few studies have focused on these noises in 3D object tracking. We add Gaussian noise to the ego-motion and 3D detection boxes to test the algorithm's robustness to noise on the Traffic Congestion dataset.

Specifically, gaussian noise with zero mean and standard deviation $\sigma$ is added to the sampled detections. We gradually increase the value of $\sigma$. Experimental results (Figure 8) show that our system maintains superior tracking results compared to other algorithms even as noise increases, demonstrating strong robustness.

## 7.4 Trajectory Evaluation

We comprehensively evaluate the trajectory accuracy on two benchmark datasets: the KITTI dataset and the Traffic Congestion dataset.

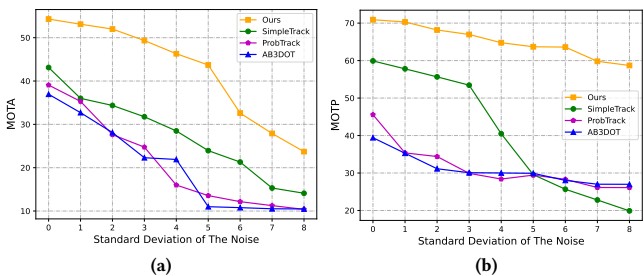

**Figure 8: MOTA(a) and MOTP(b) results in robustness evaluation experiments.**

We employ standard metrics and compare them against the state-of-the-art SLAM systems. Trajectory results are processed using the EVO tool[14]. Specifically, we consider both the Absolute Pose Error (APE) and Relative Pose Error (RPE) for each trajectory. The Root Mean Square Error (RMSE) is chosen as it provides a more robust evaluation than the mean value.

The evaluation results are detailed in Table 3 for the TCD and Table 4 for the KITTI dataset. The results show the superior performance of our approach across different datasets. For the performances on KITTI, our algorithm has achieved top-1 or top-2 in most sequences. Although the vehicles in the scenes of the KITTI dataset are not crowded, our proposed association method still achieves a modest improvement. It performs particularly better in sequences with a larger number of objects, such as in KITTI-07. In the TCD dataset, our algorithm has demonstrated significantly superior performance due to the presence of crowded objects in the scenes.

## 7.5 Ablation Experiments

We test the efficiency of the proposed modules in ablation experiments, including the Multi-criteria Star Graph Association (MSGA) and Object-centric Graph Optimization (OGO). The experiments are conducted on the proposed TCD dataset to better facilitate the evaluation of both 3D MOT and SLAM.

In the ablation experiments of OGO (Table 6), the tightly coupled OEFW significantly reduces APE, while the two-stage real-time

**Table 3: Trajectory Accuracy Evaluation on Traffic Congestion Dataset. The top one is in bold and the second is underlined.**

| TC Seq. | 0 | | 1 | | 2 | | 3 | | 4 | | Average | |
|---|---|---|---|---|---|---|---|---|---|---|---|---|
| Metrics(m) | RPE | APE | RPE | APE | RPE | APE | RPE | APE | RPE | APE | RPE | APE |
| ORBSLAM3[5] | 5.10 | 0.95 | 4.88 | 0.55 | 2.97 | 0.68 | 5.22 | 0.94 | 5.14 | 0.74 | 4.66 | 0.77 |
| DSP-SLAM[33] | 5.07 | 0.74 | 4.73 | 0.59 | 2.97 | 0.56 | 4.01 | 0.68 | 5.11 | 0.59 | 4.38 | 0.63 |
| DynaSLAM[3] | 4.86 | 0.73 | 4.83 | 0.61 | 2.98 | 0.57 | 5.2 | 0.62 | 5.09 | 0.73 | 4.59 | 0.62 |
| VDO-SLAM[42] | 4.79 | 0.69 | 4.14 | 0.63 | 2.61 | 0.59 | 4.12 | 0.65 | 4.81 | 0.50 | 4.09 | 0.61 |
| GSLAMOT(Ours) | 4.73 | 0.58 | 4.18 | 0.46 | 2.52 | 0.51 | 3.28 | 0.63 | 3.31 | 0.40 | 3.60 | 0.52 |

**Table 4: Trajectory Accuracy Evaluation on KITTI Dataset. The top one is in bold and the second is underlined.**

| KITTI Seq. | 00 | | 01 | | 02 | | 03 | | 04 | | 05 | | 06 | | 07 | | 08 | | Average | |
|---|---|---|---|---|---|---|---|---|---|---|---|---|---|---|---|---|---|---|---|---|
| Metrics(m) | RPE | APE | RPE | APE | RPE | APE | RPE | APE | RPE | APE | RPE | APE | RPE | APE | RPE | APE | RPE | APE | RPE | APE |
| ORBSLAM3[5] | 2.09 | 1.46 | 7.52 | 12.70 | 2.3 | 3.5 | 0.84 | 1.44 | 0.6 | 0.25 | 0.91 | 0.93 | 0.92 | 0.99 | 0.49 | 0.49 | 3.06 | 3.06 | 2.08 | 2.76 |
| DSP-SLAM[33] | 1.09 | 1.10 | 3.87 | 12.06 | 0.94 | 0.89 | 1.28 | 0.47 | 0.64 | 0.73 | 0.53 | 0.46 | 0.81 | 0.42 | 0.5 | 0.48 | 3.17 | 11.99 | 1.40 | 3.18 |
| DynaSLAM[3] | 1.05 | 1.28 | 3.75 | 21.13 | 1.1 | 0.91 | 0.68 | 1.43 | 0.73 | 0.82 | 0.64 | 1.52 | 0.8 | 1.35 | 0.51 | 0.78 | 3.06 | 10.41 | 1.36 | 4.40 |
| VDO-SLAM[42] | 1.02 | 1.44 | 3.80 | 13.79 | 0.98 | 0.99 | 0.79 | 0.83 | 0.61 | 0.25 | 0.59 | 0.49 | 0.75 | 0.63 | 0.49 | 0.52 | 3.34 | 9.76 | 1.50 | 3.19 |
| GSLAMOT(Ours) | 1.01 | 1.02 | 3.69 | 13.1 | 0.92 | 0.91 | 0.68 | 0.57 | 0.56 | 0.23 | 0.53 | 0.41 | 0.70 | 0.44 | 0.48 | 0.43 | 3.05 | 3.16 | 1.29 | 2.25 |

optimization of OCOW reduces RPE. For the MSGA (Table 5), spatial, neighborhood, and shape consistency all contribute to the improvement of MOT.

**Table 5: Ablation Experiments of MSGA on Traffic Congestion.**

| MSSA | MOTA↑ | MOTP↑ | APE↓ | RPE↓ |
|---|---|---|---|---|
| Spatial | 43.25 | 61.01 | 0.59 | 3.82 |
| Spatial+Neighborhood | 46.79 | 67.88 | 0.57 | 3.77 |
| Spatial+Neighborhood+Shape | 49.10 | 69.71 | 0.52 | 3.60 |

**Table 6: Ablation Experiments of OGO on Traffic Congestion.**

| OGO | MOTA↑ | MOTP↑ | APE↓ | RPE↓ |
|---|---|---|---|---|
| OEFW | 43.66 | 59.91 | 0.63 | 4.21 |
| OCOW | 48.93 | 63.40 | 1.30 | 3.81 |
| OCOW+OEFW | 49.10 | 69.71 | 0.52 | 3.60 |

## 7.6 Time Evaluation

To evaluate the system's efficiency and real-time applicability, we measure the average processing time per frame, encompassing all stages, including ego-motion pose estimation, object analysis, association, optimization, and map integration. We conduct experiments over sequences from the KITTI and TCD. The average computation time is reported. The evaluation results in Table 7 affirm GSLAMOT's real-time capability and computational efficiency.

We evaluate the running time of each main module in the system. Note that some modules, such as detection, tracking, and optimization, run only on keyframes selected by the VO front-end. To facilitate a more intuitive time comparison, we calculate the average time expenditure of all modules across all frames. The results are shown in Table 8. Due to the parallel computing strategy, the total time expenditure per frame is significantly less than the time consumed by serial computation of individual modules.

**Table 7: Time Evaluation.**

| Method | Time Per Frame(s)↓ |
|---|---|
| ORBSLAM3[5] | 0.025 |
| DSP-SLAM[33] | 0.12 |
| DynaSLAM[3] | 0.086 |
| VDO-SLAM[42] | 0.14 |
| GSLAMOT( ours) | 0.082 |

**Table 8: Running Time of the Main Modules in Our System.**

| Module | Time Per Frame(s)↓ |
|---|---|
| Ego-motion Tracking | 0.031 |
| Mapping | 0.027 |
| 3D Detection† | 0.022 |
| MSSA† | 0.024 |
| OGO† | 0.037 |
| Total | 0.082 |

†: only for keyframes.

## 8 CONCLUSION

This paper investigates the problem of Simultaneous Locating, Mapping, and 3D Multiple Object Tracking. We propose the MSGA algorithm, which computes features from query graphs and tracklet graphs to achieve detection and tracklet association. After association, we design the OGO strategy to optimize ego-motion and tracklet poses in multiple stages. Additionally, we create a simulation dataset to explore the challenges of 3D MOT in highly congested, dynamic scenes. In the future, we plan to extend GSLAMOT to multi-agent collaborative scenarios to leverage the information from multiple agents and viewpoints to enhance system performance.

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
