# OpenReview forum: "GSLAMOT: A Tracklet and Query Graph-based Simultaneous Locating, Mapping, and Multiple Object Tracking System"
_acmmm.org/ACMMM/2024/Conference — MM2024 Poster_

### Official Review · Reviewer_QwmW · 2024-05-24

**Rating:** 4
**Confidence:** 1

**Summary:**

The work presents a graph matching and graph optimization-based method for performing SLAM (self-locating) and 3D MOT (tracking) jointly providing both ego trajectory and object trajectories even in crowded and highly dynamic scenes. Moreover, the method outputs also mapping results together with other outputs. This makes the proposed method a locating, mapping, and MOT system.

**Strengths:**

- The idea of jointly performing localization, tracking, and mapping is novel. The work shows competitive results even in challenging scenarios like crowded scenes.
- A multi-criteria star graph association is proposed to match different modalities.
- The work also addresses the runtime issue considering the multi-task nature of the system.

**Limitations:**

- The work is validated on a self-created dataset based on a possible unrealistic simulator. A performance drop is observed in real-world scenes like the Waymo Open dataset. More discussion on the performance gap can be further addressed.
- The used motion model in Line 291 can be further detailed.
- The dependency of the object detector is not addressed while the proposed method depends on the performance of the used object detector.

**Suitability:**

2

---

### Official Review · Reviewer_KvjG · 2024-05-24

**Rating:** 4
**Confidence:** 1

**Summary:**

The paper introduces GSLAMOT, designed for the tasks of locating, mapping, and tracking the 3D poses of multiple objects in unfamiliar settings. It leverages both camera and LiDAR multimodal data, employing a Tracklet and Query Graph-based framework to effectively handle dynamic scenes and achieve precise object association. Results indicate that the proposed method outperforms state-of-the-art techniques in accurately tracking crowded objects and conducting SLAM in difficult conditions.

**Strengths:**

1. GSLAMOT can effectively leverages multimodal information and employs graph matching and optimization for simultaneous SLAM and 3DMOT even in challenging scenarios.
2. The  Multi-criteria Star Graph Association (MSGA) and Object-Centric Graph Optimization (OGO) improve object matching and pose estimation. Object-Ego Fusion Window (OEFW) optimizes object and ego motion poses in a long-term sliding window, enhancing overall performance. The authors validated the effectiveness of these modules in the experiments.
3. The article's textual descriptions are clear.

**Limitations:**

1. In Section 4.1, the authors obtain object tracklets using traditional methods. Would replacing this with a deep learning-based tracking methods work better?
2. In the speed testing experiment in Section 7.6, the authors should specify the hardware conditions used for testing.
3. The author's page count does not fill the last page completely; the length should be appropriately adjusted to fill the final page.

**Suitability:**

3

---

### Official Review · Reviewer_1NED · 2024-05-24

**Rating:** 3
**Confidence:** 2

**Summary:**

The paper proposes GSLAMOT to simultaneously locate, map, and track the 3D poses of multiple moving objects. The framework utilizes camera and LiDAR information to represent dynamic scenes through semantic maps, proxy tracks, and tracklet graphs maintained online. It uses a multi-standard star graph association method and an object-centered graph optimization method to accurately correlate and optimize objects, semantic maps, and proxy trajectories. Experimental results show that GSLAMOT performs better than existing methods.

**Strengths:**

1.This method can output ego trajectory and object trajectories concurrently, which is novel.
2.The experiment is relatively sufficient.
3.A new dataset proposed, which is focused on scene with dense and dynamic flow.

**Limitations:**

1.Does the method make a fair comparison with other methods on those existing datasets, e.g., fair inputs, fair data, fair pre-trained models?
2.TrajectoryFormer[6] have significantly higher results that have not been reported on the Waymo Open Dataset.
3.The objective function and errors are not clear in Sec.5. What parameters is trained? Why can such a design achieve the effect claimed by the author?
4.The details of the proposed dataset are neglected. And, I think this dataset might be of more interest to the community. I think the contribution of the paper needs to be reorganized.

**Suitability:**

2

---

### Official Review · Reviewer_1gLG · 2024-05-25

**Rating:** 4
**Confidence:** 2

**Summary:**

This paper focuses on ego trajectory and object trajectory concurrently for the first time. Specifically, it first proposes Multi-criteria Star Graph Associatoin to accurately
match query graph and tracklet graph based on neighborhood consistency, spatial consistency and shape consistency.
Then, it utilizes Object-centric Optimization Window  and Object-Ego Fusion Window from the respect of object detection and ego-motion for further optimization.

**Strengths:**

1.This paper associates two independent tasks (i.e., Object-oriented SLAM and 3D Multiple Object Tracking) for the first time and achieves outstanding performance.
2. The experimental results are relatively superior on widely used KITTI benchmark and Waymo Open Dataset.
3. The structure of this paper is good and it is easy to follow this paper.

**Limitations:**

1. Section 7.3 Robustness Evaluation only proves that the proposed GSLAMOT has higher tracking accuracy compared to other methods with Gaussian detection noise, which may be related to your method combining ego-motion and object detection to make tracking predictions. I think a specialized ablation experiment needs to be designed to demonstrate that GSLAMOT can prevent interference from detection errors (Section 5).
2. Section 7.2 3D Object Tracking Evaluation lacks of comparison on KITTI which is also commonly used on 3D MOT with available point cloud and stereo images.
3. Paper Writing has tense inconsistency. For example, in line 641 and 665, collected -> collect, evaluated -> evaluate

**Suitability:**

2

---

### Meta-Review · Area_Chair_gQVc · 2024-06-29

**Recommendation:** Accept (Poster)
**Confidence:** 5

**Metareview:**

All four reviewers are positive about the work and aprreciated the technqiue contribution and experimental validation in general. Some concerns were raised and were late mostly addressed by the authors' rebuttal. The final ratings are borderline accepts or weak accept. After checking all the reviews and rebuttal, we agree with the reviwer team and would like to recommend the paper to ACMMM.